# Towards Robust Uncertainty Calibration for Composed Image Retrieval

**Yifan Wang**[1], **Wuliang Huang**[2], **Yufan Wen**[1], **Shunning Liu**[1], **Chun Yuan**[1]*

[1]Tsinghua Shenzhen International Graduate School, Tsinghua University

[2]Institute of Computing Technology, Chinese Academy of Sciences

{yifan-wa22@mails, wenyf24@mails, lsn24@mails, yuanc@sz}.tsinghua.edu.cn

huangwuliang19b@ict.ac.cn

## Abstract

The interactive task of composed image retrieval aims to retrieve the most relevant images with the bi-modal query, consisting of a reference image and a modification sentence. Despite significant efforts to bridge the heterogeneous gap within the bi-modal query and leverage contrastive learning to reduce the disparity between positive and negative triplets, prior methods often fail to ensure reliable matching due to aleatoric and epistemic uncertainty. Specifically, the aleatoric uncertainty stems from underlying semantic correlations within candidate instances and annotation noise, and the epistemic uncertainty is usually caused by overconfidence in dominant semantic categories. In this paper, we propose *Robust UNcertainty Calibration* (**RUNC**) to quantify the uncertainty and calibrate the imbalanced semantic distribution. To mitigate semantic ambiguity in similarity distribution between fusion queries and targets, RUNC maximizes the matching evidence by utilizing a high-order conjugate prior distribution to fit the semantic covariances in candidate samples. With the estimated uncertainty coefficient of each candidate, the target distribution is calibrated to encourage balanced semantic alignment. Additionally, we minimize the ambiguity in the fusion evidence when forming the unified query by incorporating orthogonal constraints on explicit textual embeddings and implicit queries, to reduce the representation redundancy. Extensive experiments and ablation analysis on benchmark datasets FashionIQ and CIRR verify the robustness of RUNC in predicting reliable retrieval results from a large image gallery.

## 1 Introduction

The task of composed image retrieval (CIR) [1, 2, 3, 4, 5] is emerging as a multi-modal interaction form to accommodate the flexible search requirements. Distinguished from traditional single-modal image retrieval or cross-modal retrieval, the queries of CIR support two modalities, *i.e.*, reference images and modification texts that illustrate alternations on the reference images. To identify the most correlated images among the massive candidate images, the core objective is to establish semantic connections between target images and bi-modal queries through similarity measurement and bridge the heterogeneous modality gap across different modalities within queries. Exploring multi-modal data integration and natural interaction requirements on this task could provide underlying support for tasks such as visual question answering (VQA) [6, 7, 8], visual reasoning [9, 10, 11], etc. as the basis of multi-modal understanding [12].

One widely recognized challenge for composed image retrieval is the comprehension of the semantic conflict in the bi-modal query. Despite the modality gap in the query composition, the modifications

---

*Corresponding Author

39th Conference on Neural Information Processing Systems (NeurIPS 2025).

expressed by the query text have led to disagreements with the reference images, hindering the understanding of the multi-modal input and formation of a unified query representation. Existing

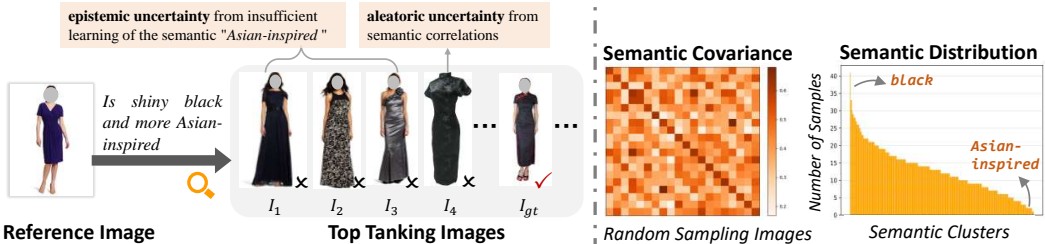

Figure 1: Illustration of uncertain matching results. (*left*) Unreliable top-ranking results disrupted by insufficient learning of "*Asign-inspired* " and partial semantic correlations. (*right*) Strong correlations within the semantic matrix and imbalanced semantic distributions underlying visual candidates.

works were devoted to mining the effective components within the queries by multi-granular feature matching, *e.g.*, word-level [13, 14], patch-level [15], token-level [16, 17], and hierarchical-level [18]. Inspired by the significant achievement of vision-language models on the massive corpus, recent models adopted transformer-based encoders in CLIP [19, 20] and BLIP [21, 22] to enhance visual and textual features [23] and align semantics in the query, owing to multiple inter- and intra-attention mechanisms to extract salient information. Besides, to capture the correspondences between multi-modal queries and targets, current approaches [24, 5] mainly project the query and target features in the joint space and measure pairwise similarities in the contrastive learning framework. This pipeline regards the CIR task as a classification task to distinguish between matched and mismatched triplets.

Contrastive learning essentially follows the assumption that there is no overlap between the distributions of positive and negative triplets. However, images inherently contain rich semantic concepts that cannot be explicitly assigned as independent labels, *i.e.*, negative candidates exhibit certain underlying semantic dependencies with ground-truth positive images. Fig. 1 illustrates that the annotated "negative" images $I_4$ and ground-truth images $I_{gt}$ exhibit strong visual relations, while the semantic correlations remain disregarded within contrastive learning paradigm. Moreover, it is inevitable that not all images that satisfy the query requirements are labeled as positives, and the modification descriptions lack clarity in accurately conveying the intended image. The collaborative impacts of the above factors lead to *aleatoric uncertainty*, which is caused by the implicit semantics dependencies and noise labelling issues in the datasets. Additionally, semantic concept imbalance is also significant in images where specialized designs like "Asian-inspired" are limited in the fashion domain, as seen in Fig. 1 (*right*). Furthermore, statistical properties of widely-used softmax function in either attention mechanisms [25, 13, 23, 21] or cross-entropy calculation, further increase the biased estimation, leading to overconfident predictions dominated by the explicit semantic concepts and overlook of discriminative details (as shown in $I_1$, $I_2$, and $I_3$ in Fig. 1). The imbalanced distributions of semantic concepts in visual features and over-concentration on salient features result in *epistemic uncertainty*, which tends to overfit the majority semantics and result in low confidence when facing minority semantic categories.

To address the aforementioned challenges, we propose *Robust UNcertainty Calibration* (RUNC) to ensure credible semantic bridging between bi-modal queries and visual targets, as shown in Fig. 2. In order to perceive the latent semantic correlations and quantify uncertainty, we incorporate Normal Inverse Gamma distribution as evidential priors to fit the semantic covariances in the candidate images. Through maximizing the evidence by the model and imposing a penalty on the incorrect evidence, the inferred probabilistic distributions estimate aleatoric and epistemic uncertainty on each candidate image. With more emphasis on uncertain images with rare and ambiguous semantics, we assign uncertainty coefficients to calibrate the target distribution when supervising the query distribution. To further decrease the ambiguity when composing fusion representations for the coupled query images and texts and ensure the semantic consistency, implicit query embedding is introduced to drive the query embeddings to distill more retained visual semantics rather than redundant representations in text modifiers through aligning with the targets and orthogonal to query texts. Experimental results on widely-adopted datasets FashionIQ and CIRR verify that RUNC yields robust and reliable rankings.

In summary, the proposed RUNC makes the following contribution:

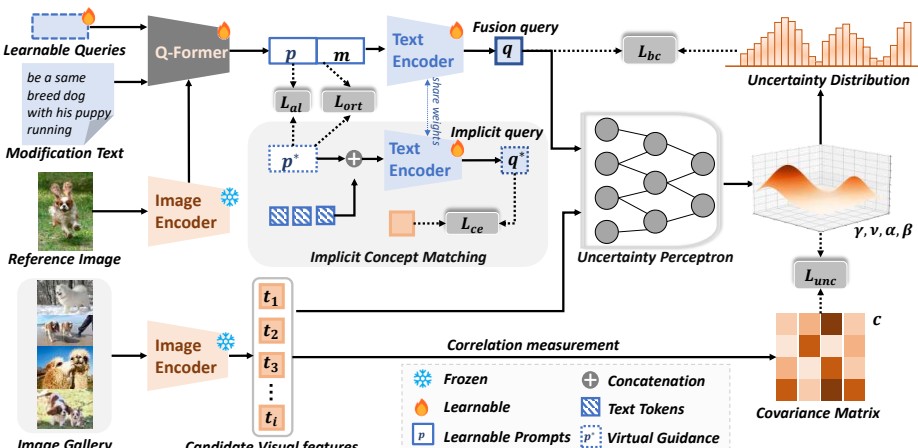

Figure 2: **The Framework of RUNC**. Uncertainty perceptron introduces evidential priors to fit the semantic covariance and yield uncertainty distribution to calibrate the supervision on the fusion query. Implicit guidances $p^*$ are incorporated to distill effective features for retention and modification.

- We propose a novel uncertainty calibration approach to address the misguidance of dominant semantics and semantic overlaps when matching bi-modal queries with target images.
- We employ high-order evidential priors to quantify the aleatoric and epistemic uncertainty and adaptively adjust the semantic distribution imbalance based on uncertainty estimation.
- To minimize the ambiguity when fusing hybrid-modal queries, we assign orthogonal independent constraints on explicit textual embeddings and implicit queries to distill effective features for retention and modification.

## 2 Related Works

### 2.1 Composed Image Retrieval

To address the growing need for flexible retrieving with multi-modal data, composed image retrieval focuses on exploring the integration of dual-modal input of reference images and modification text and matching with the desired images. Existing works are dedicated to mitigating the heterogeneous gap of multi-modal data in the query domain by extracting effective semantic components across different modalities [1, 26, 27] via multiple projection layers [1], cross-modal attention [18, 28], and graph-based propagation [13, 29]. For instance, FashionVLP [30] introduced multi-layer self-attention on the combined tokens of visual regions from the reference image and words from modifiers. To facilitate correlations with query images and query texts, Bai *et al.* [31] incorporated sentence-level prompts with visual features and textual tokens in the Q-Former structure [32]. Through contrastive learning, the fusion queries are guided toward the target features and pushed away from other candidate features by comparing positive and negative samples. DCNet [33], ComposeAE [27], and CaLa [22] applied bi-directional constraints to strengthen the semantic consistency across references, modifiers, and targets. Recent advances [34, 16] reconstructed triplet data with higher-quality to ensure that contrastive learning effectively captures semantic alignment, which requires expensive annotation efforts. With a focus on the separation of positive and negative matching in the above approaches, the latent semantic correlations among the images, particularly false negatives, may hinder the actual semantic alignment. Distinguished from previous methods, this work deploys prior distributions to model the intrinsic covariance of candidate features and addresses the semantic imbalance by uncertainty calibration to enhance the robustness.

### 2.2 Uncertainty Estimation

Though deep learning models are currently excelling across various domains, most of them typically provide predictions without considering the confidence of the outcomes [35, 36]. The prediction uncertainty stemming from data noise, model overconfidence, and biased learning can significantly

impact the robustness of noisy labels, generalization on unseen classes, and model interpretability. Based on Bayesian Neural Networks (BNN) [37] and Monte Carlo Dropout, uncertainty models [38] estimated prediction variance through multiple forward propagation samples in out-of-distribution (OOD) scenarios [39, 40]. MGUR [4] applied weighted Gaussian noises on whitened features to simulate data jitter and one-to-many correspondences. Subjective Logic (SL) formalized the concept of belief assignment in the Dempster-Shafer theory of evidence as a Dirichlet distribution [41, 42] to quantify the belief mass and uncertainty. Deep Evidential Regression (DER) [43, 44] introduced high-order priors to capture the confidences supporting the model prediction. Evidential deep learning avoided the computational bottlenecks of traditional Bayesian methods by obtaining uncertainty in a single forward inference. In this work, the proposed RUNC extends the uncertainty estimation to explore latent correlations in the coupling visual semantics and calibrate the imbalanced semantic distributions when aligning composed queries and targets.

## 3 Methodology

### 3.1 Problem Setting

The multi-modal dataset for composed image retrieval includes $N$ query-to-target pairs. Each matched data $(\mathcal{I}_r, M, \mathcal{I}_t)$ contains one reference image $\mathcal{I}_r$, one modification sentence $\mathcal{M}$, and one target image $\mathcal{I}_t$. To bridge the modality gap between the visual and textual inputs, the retrieval pipeline utilizes pretrained encoders [19, 21] to project all the raw image inputs into the semantic latent space. We denote candidate visual features as $\{t_i\}_{i=1}^N$ for $N$ images in the gallery, where $t_i \in \mathbb{R}^d$ and $d$ denotes dimensions. In the branch of combining the cross-modal query, the lightweight Q-Former [32] is employed to obtain the interactive prompt embedding $p$ along with text features $m$, with the input of visual features from the original reference images, instructions from the modification text, and learnable queries. Afterward, the prompt embedding $p$ is further projected in the same latent space as $t$ to yield the fusion query representation $q$.

The essence of the composed image retrieval task lies in accurately measuring the semantic distances between the queries and targets in the embedding space. With the aim of pushing the fusion query representations towards the target features ($q_i \rightarrow t_i$) while separating query representations from irrelevant candidate samples ($q_i \longleftrightarrow t_j$), existing frameworks [45, 21] mainly adopt the contrastive loss to classify positive pairs ($q_i, t_i$) and negative pairs ($q_i, t_j | j \neq i$):

$$\mathcal{L}_{cl} = -\frac{1}{B} \sum_{i=1}^B \log \frac{\exp(s_{ii}/\tau)}{\sum_{j=1}^B \exp(s_{ij}/\tau)}, \tag{1}$$

where $B$ is the batch size and $\tau$ refers to the temperature parameter. The matching score $s_{ij}$ between composed query $i$ and candidate feature $j$ is computed based on the cosine distance $s_{ij} = \frac{q_i \cdot t_j}{\|q_i\|\|t_j\|}$, where $\|\cdot\|$ means the L2 normalization.

Though cross-entropy loss is efficient in large-scale retrieval applications, it cannot be fully adapted to this complicated interactive retrieval task. In this context, contrastive loss is primarily focused on aligning the matching scores matrix $s \in \mathbb{R}^{B \times B}$ with the diagonal matrix. Specifically, for normalized features, the matching value for the fusion query to the target features should ideally approach 1, while the matching value for all the other samples should be as close to 0 as possible. However, due to the strong semantic correlations among most candidate images in the image gallery and noise labels during the dataset construction process, non-diagonal negative samples may exhibit semantic overlap. Furthermore, the fused features derived from the complex multi-modal interactive inputs inherently introduce semantic uncertainty and instability, which results in loose and unstable semantic construction with traditional supervision loss. To this end, the uncertainty of alignments between the cross-modal query and candidate images is imperative to estimate to construct a robust retrieval model. As shown in Fig. 2, we deploy distribution-based uncertainty estimation to quantify the semantic ambiguity and bring aleatoric and epistemic uncertainty into consideration when setting the training objective.

### 3.2 Uncertainty Estimation

**Priors of Semantic Covariance Matrix.** As aforementioned, the high frequency of coupling semantic concepts (as shown in Figure 1 (*left*)) in the candidate images poses challenges to the

model's prediction of similarities between queries and targets. Using hard-coded supervision with binary values of 1 or 0 for positive and negative sample pairs respectively fails to accurately represent the real retrieval scenario. In the proposed RUNC, we approach the prediction of matching values in the retrieval process as a regression problem, assuming that the distribution of the semantic correlations conforms to Gaussian distributions and is independent and identically distributed (i.i.d). The correlations across various target semantic concepts are acquired by the interactions of candidate features in the target image space, to measure the potential semantic overlaps between different pairs.

$$c_{ij} = \frac{t_i \cdot t_j}{\|t_i\|\|t_j\|}, \tag{2}$$

where $c_{ij}$ stands for the semantic covariance score between the $i$-th sample and the $j$-th sample.

The objective of uncertainty estimation is to estimate a prior distribution to reconstruct the variance and mean of the Gaussian distribution of semantic concepts [46, 47, 43]. Therefore, a high-order Normal Inverse Gamma (NIG) prior is introduced to model the Gaussian output:

$$(c_{1i}, c_{2i}, ..., c_{Ni}) \sim \mathcal{N}(\mu_i, \delta_i^2), \quad \mu_i \sim \mathcal{N}(\gamma_i, \delta_i^2 \nu_i^{-1}), \quad \delta_i^2 \sim \Gamma^{-1}(\alpha_i, \beta_i), \tag{3}$$

where $\Gamma(\cdot)$ represents Gamma function.

To estimate the posterior distribution $q(\mu_i, \delta_i^2) = p(\mu_i, \delta_i^2 | \gamma_i, \nu_i, \alpha_i, \beta_i)$ for the $i$-th target image semantic representations, we factorize the distribution in the form of conjugate prior as $q(\mu_i, \delta_i^2) = q(\mu_i)q(\delta_i^2)$, which is reformulated as:

$$p(\mu_i, \delta_i^2 | \gamma_i, \nu_i, \alpha_i, \beta_i) = \frac{\beta_i^{\alpha_i}\sqrt{\nu_i}}{\Gamma(\alpha_i)\sqrt{2\pi\delta_i^2}}(\frac{1}{\delta_i^2})^{\alpha_i+1}\exp\{-\frac{2\beta_i + \nu_i(\gamma_i - \mu_i)^2}{2\delta_i^2}\}. \tag{4}$$

Given the fusion query representations and the evidential distribution from uncertainty perceptron, we then maximize the model evidence to support the observations by maximizing the likelihood of observing the semantic covariance matrix:

$$\begin{aligned}\mathcal{L}_i^{NLL} &= -\log(p(c_{ij}|\gamma_i, \nu_i, \alpha_i, \beta_i)) \\ &= \frac{1}{2}\log(\frac{\pi}{\nu_i}) - \alpha_i\log\Omega_i + (\alpha_i + \frac{1}{2})\log((c_{ij} - \mu_i)^2\nu_i + \Omega_i) + \log(\frac{\Gamma(\alpha_i)}{\Gamma(\alpha_i + \frac{1}{2})}),\end{aligned} \tag{5}$$

where $\Omega_i = 2\beta_i(1 + \nu_i)$. Compared to directly imposing the hard-encoded labels on the distances between the fusion query and target, introducing evidential distribution to capture semantic interactions between various instances facilitates reliable similarity measurements in a more nuanced way.

**Uncertainty-Guided Semantic Calibration.** From the perspective of Bayesian Inference, NIG distribution is a conjugate prior of Gaussian distribution, and its corresponding parameter could be intuitively interpreted as virtual observations. To reveal the shape characteristics, the mean could be regarded as the sample mean calculated from $\nu$ virtual observations with the mean of $\gamma$, and the variance could be considered as an estimation based on $\alpha$ virtual observations with the mean of $\gamma$ and the sum of squared deviations $2\nu$. Thus, the total evidence, which is the sum of all the inferred virtual observations comprised of all the virtual observation information of means and variances, is defined as $2\nu + \alpha$. Based on the model parameters of the uncertainty model in Section 3.2, the statistical moments of target semantics are computed through first-order moments of the NIG distribution:

$$\mathbb{E}[\mu] = \gamma, \quad \mathbb{E}[\delta^2] = \frac{\beta}{\alpha - 1}, \quad \text{Var}[\mu] = \frac{\beta}{\nu(\alpha - 1)}, \tag{6}$$

where the latter two notions could also be interpreted as aleatoric and epistemic uncertainty of semantic distribution. The total uncertainty is further obtained by:

$$u_i = \mathbb{E}[\delta_i^2] + \text{Var}[\mu_i] = \frac{\beta_i}{\alpha_i - 1} + \frac{\beta_i}{\nu_i(\alpha_i - 1)} = \frac{\beta_i(\nu_i + 1)}{\nu_i(\alpha_i - 1)}. \tag{7}$$

Note that semantic distribution is imbalanced in the realistic retrieving process. For instance, semantics related to colors and objects tend to appear frequently, while those involving specific details like "whimsical and vintage" are rare. Consequently, the top retrieval results tend to predict semantic categories that are more commonly represented. To avoid imbalanced learning from diverse semantics, we introduce uncertainty coefficients on cross-entropy computation based on the semantic

uncertainty fitted by NIG distribution. When a semantic embodies high uncertainty, the corresponding weight is ought to set larger, so that more penalty would be enforced on this sample during the training phase, avoiding neglecting infrequent semantic categories. The refined balanced contrastive loss is:

$$\mathcal{L}_{bc} = -\frac{1}{B}\sum_{i=1}^{B} u_i \cdot \log \frac{\exp(\boldsymbol{s}_{ii}/\tau)}{\sum_{j=1}^{B} \exp(\boldsymbol{s}_{ij}/\tau)}. \tag{8}$$

**Penalty on Misleading Evidence.** As the likelihood-based loss maximizes the evidence, it may lead to evidence magnification associated with the target, whereas the model discrimination on matched and hard negative samples would be limited. In particular, explicit semantics tend to overpower the outcomes and mislead the model training, whereas those with limited occurrences could be overlooked during this process. Since the misleading evidence of dominant semantics could be effective in most cases, the model is prone to maintain this incorrect evidence. It is contradictory to our target to actively reveal uncertainty when dealing with ambiguous decision boundaries instead of giving wrong predictions. Thus, we introduce the regularization term on misleading evidence to ensure robust ranking from reliable evidence. Note that $\mathbb{E}[\delta^2]$ and $\text{Var}[\mu]$ in Eq. 6 both demonstrate that uncertainty shows positive correlations with the parameter $\beta$ and evidence by virtual observation theory corresponds to $\nu$ and $\alpha$. Theoretical analysis is illustrated in the supplementary material. Hence, the regularization term is defined as:

$$\mathcal{L}_i^{REG} = (\mathbf{c}_{ij} - \gamma_i)^2 \cdot (2\nu_i + \alpha_i + \frac{1}{\beta_i}). \tag{9}$$

The overall uncertainty loss combines the likelihood function to maximize the model evidence and penalty for misleading evidence with controllable weight $\lambda_1$:

$$\mathcal{L}_{unc} = \frac{1}{B}\sum_{i}^{B}(\mathcal{L}_i^{NLL} + \lambda_1 \mathcal{L}_i^{REG}). \tag{10}$$

## 3.3 Implicit Concept Matching

Apart from uncertainty matching caused by underlying semantic covariance and distribution imbalance, complicated information sources from different modalities in the hybrid-modal query also introduce ambiguity when composing the fusion query. The reference image contains redundant visual information, *e.g.*, objects and attributes that would be replaced in the modification sentences, yet these salient features are inclined to be amplified through attention layers in the transformer-based Q-Former structure, significantly corrupting the semantic representation of the fusion query by substantial noise. Moreover, learnable prompts $\boldsymbol{p}$ involved in the computation of fusion query attempts to capture visual semantics aligned with text, however the learnable prompts $\boldsymbol{p}$ may have unclear concepts due to the misleading visual redundancy. Simply equipping supervision between fusion queries and candidates could result in substantial redundant information in embedding $\boldsymbol{p}$, especially repeatedly expressing semantics already revealed by text embeddings of modifiers.

In the training phase, we additionally integrate virtual guidance $\boldsymbol{p}^*$ to directly lead the learning of prompt and fully unleash the potentials of $\boldsymbol{p}$. MSE loss is utilized to align the learnable embeddings with virtual guidance $\boldsymbol{p}^*$ as $\mathcal{L}_{al} = \|\boldsymbol{p} - \boldsymbol{p}^*\|$. To encourage the learnable queries to acquire highly correlated messages with the targets, we deploy a symmetrical representation $\boldsymbol{q}^*$ to mirror the fusion query $\boldsymbol{q}$, as shown in Fig. 2, which combines virtual guidance $\boldsymbol{p}^*$ and text tokens of modifiers. After encoding by the text encoder using shared weights, we yield the implicit query $\boldsymbol{q}^*$, which is expected to be aligned with the target features:

$$\mathcal{L}_{ce} = -\frac{1}{B}\sum_{i=1}^{B} \log \frac{\exp(s_{ii}^*/\tau)}{\sum_{j=1}^{B} \exp(s_{ij}^*/\tau)}, \tag{11}$$

where $s_{ij}^*$ is the cosine similarity between the implicit query $\boldsymbol{q}^*$ and target images.

In order to force the learnable embeddings to grasp implicit semantics in reference images rather than duplicated semantics mentioned in the modification text, we incorporate orthogonal loss to differentiate the retention and modification characteristics:

$$\mathcal{L}_{ort} = \sum_{i \neq j}(\text{Cov}(\boldsymbol{p}^*, \boldsymbol{m})_{ij})^2 = \sum_{i \neq j}(\frac{1}{B}((\boldsymbol{p}^*)^\top \boldsymbol{m})_{ij})^2. \tag{12}$$

The overall training objective is the aggregation of all the loss functions as: $\mathcal{L} = \mathcal{L}_{bc} + \lambda_2 \mathcal{L}_{unc} + \mathcal{L}_{ort} + \mathcal{L}_{al} + \mathcal{L}_{ce}$, where $\lambda_2$ is a trade-off parameter. The first two terms are dedicated to quantifying the uncertainty by NIG priors and calibrating the imbalanced correlations, and the remaining terms provide soft supervision from the implicit query to minimize fusion ambiguity.

Table 1: Retrieval results on FashionIQ. The best results are marked in **bold**.

| Methods | Dress | | Shirt | | Toptee | | Average | | |
|---|---|---|---|---|---|---|---|---|---|
| | R@10 | R@50 | R@10 | R@50 | R@10 | R@50 | R@10 | R@50 | Mean |
| VAL [18] | 21.12 | 42.19 | 21.03 | 43.44 | 25.64 | 49.49 | 22.60 | 45.04 | 33.82 |
| CIRPLANT [48] | 14.38 | 34.66 | 13.64 | 33.56 | 16.44 | 38.34 | 14.82 | 35.52 | 25.17 |
| CoSMo [49] | 21.39 | 44.45 | 16.90 | 37.49 | 21.32 | 46.02 | 19.87 | 42.65 | 31.26 |
| CLVC-Net [50] | 29.85 | 56.47 | 28.75 | 54.76 | 33.50 | 64.00 | 30.70 | 58.41 | 44.56 |
| ARTEMIS [24] | 27.16 | 52.40 | 21.78 | 43.64 | 29.20 | 54.83 | 26.05 | 50.29 | 38.17 |
| FashionVLP [30] | 26.77 | 53.20 | 22.67 | 46.22 | 28.51 | 57.47 | 25.98 | 52.30 | 39.14 |
| NSFSE [28] | 31.12 | 55.73 | 24.58 | 45.85 | 31.93 | 58.37 | 29.17 | 53.24 | 41.26 |
| CLIP4Cir [45] | 31.63 | 56.67 | 36.36 | 58.00 | 38.19 | 62.42 | 35.39 | 59.03 | 47.21 |
| CRN [51] | 30.20 | 57.15 | 29.17 | 55.03 | 33.70 | 63.91 | 31.02 | 58.70 | 44.86 |
| MGUR [4] | 32.61 | 61.34 | 33.23 | 62.55 | 41.40 | 72.51 | 35.75 | 65.47 | 50.61 |
| SPN [34] | 38.82 | 62.92 | 45.83 | 66.44 | 48.80 | 71.29 | 44.48 | 66.88 | 55.68 |
| FAME-ViL [52] | 42.19 | 67.38 | 47.64 | 68.79 | 50.69 | 73.07 | 46.84 | 69.75 | 58.29 |
| CaLa [22] | 42.38 | 66.08 | 46.76 | 68.16 | 50.93 | 73.42 | 46.69 | 69.22 | 58.05 |
| SPRC [31] | 49.18 | 72.43 | 55.64 | 73.89 | 59.35 | 78.58 | 54.92 | 74.97 | 64.85 |
| CCIN [53] | **49.38** | 72.58 | 55.93 | 74.14 | 57.93 | 77.56 | 54.41 | 74.76 | 64.59 |
| **RUNC (Ours)** | 48.93 | **73.53** | **57.26** | **75.32** | **60.38** | **79.86** | **55.52** | **76.23** | **65.88** |

## 4 Experiments

### 4.1 Experimental Settings

**Datasets.** The proposed RUNC is employed on two widely-used composed image retrieval datasets covering various modification requirements in real-life retrieval scenarios. **FashionIQ** [54], concentrating on fashion item retrieval, addresses the retrieval for modifications in attributes including colors, patterns, textures, and design details across dress, toptee, and shirt categories. The whole dataset contains 77,684 fashion pictures and each matched triplet is constituted of a reference image, a modification sentence, and one target image. Following [1, 18], the dataset is split by the proportion of 3:1:1 for training, validating, and testing respectively. **CIRR** [48] involves more natural scenes and query texts more focus on alterations in the relationships among subjects, backgrounds, and multiple subjects within intricate images. It consists of 21,552 images collected from the NLVR$^2$ dataset [55] and constructs 36,554 matched pairs. To further evaluate the model when facing different scenarios, CIRR also provides a subset setting and each subset includes six visually similar images.

**Evaluation Metrics.** Following [24, 34, 31], we employ the Recall rate at $K$ (R@$K$) as the main metric to evaluate the model performance, which is defined as the ratio of matched ground-truth images ranked in the top-$K$ predictions by the model. In FashionIQ, R@10 and R@50 results are shown on dresses, toptees, and shirts. In CIRR, apart from R@1, R@5, R@10, and R@50 metrics, we also provide $R_{subset}@K$ results evaluated in the subsets.

**Implementation Details.** We exploited the visual and textual encoders as BLIP-2$_{\text{ViT}-\text{G}/14}$ model and initialized parameters from pre-trained EVA-CLIP [32] weights. The visual encoder remained frozen and the remaining layers were fine-tuned in the training phase. The virtual guidance was disabled during the inference phase. The uncertainty perceptron was implemented as one feed-forward network (two linear layers) with a softplus activation function. The dropout rate was set as 0.2. The dimensions of fusion and candidate features were fixed as 256 in the embedding space and the number of learnable queries was set as 32. We set $\lambda_1$ as 0.01 to in Eq. 10. We used AdamW optimizer and set the learning rate as $2 \times 10^{-5}$ in FashionIQ and $1 \times 10^{-5}$ in CIRR with cosine annealing decay. The training and inference time of the proposed model are 214.3s and 27.4s respectively. The

Table 2: Retrieval results on CIRR test set. The best results are marked in **bold**.

| Methods | Recall@$K$ | | | | $R_{subset}@K$ | | | Avg(R@5, $R_{subset}@1$) |
|---|---|---|---|---|---|---|---|---|
| | K=1 | K=5 | K=10 | K=50 | K=1 | K=2 | K=3 | |
| TIRG [1] | 11.04 | 35.08 | 51.27 | 83.29 | 23.82 | 45.65 | 64.55 | 29.45 |
| CIRPLANT [48] | 19.55 | 52.55 | 68.39 | 92.38 | 39.20 | 63.03 | 79.49 | 45.88 |
| ARTEMIS [24] | 16.96 | 46.10 | 61.31 | 87.73 | 39.99 | 62.20 | 75.67 | 43.05 |
| NSFSE [28] | 20.70 | 52.50 | 67.96 | 90.74 | 44.20 | 65.53 | 78.50 | 48.35 |
| CLIP4Cir [45] | 33.59 | 65.35 | 77.35 | 95.21 | 62.39 | 81.81 | 92.02 | 63.87 |
| CompoDiff [56] | 22.35 | 54.36 | 73.41 | 91.77 | 35.84 | 56.11 | 76.60 | 45.10 |
| BLIP4CIR [21] | 40.17 | 71.81 | 83.18 | 95.69 | 72.34 | 88.70 | 95.23 | 72.07 |
| SSN [57] | 43.91 | 77.25 | 86.48 | 97.45 | 71.76 | 88.63 | 95.54 | 74.51 |
| SPN [34] | 45.33 | 78.07 | 87.61 | 98.17 | 73.93 | 89.28 | 95.61 | 76.00 |
| CaLa [22] | 49.11 | 81.21 | 89.59 | 98.00 | 76.27 | 91.04 | 96.46 | 78.74 |
| SPRC [31] | 51.96 | 82.12 | 89.74 | 97.69 | 80.65 | 92.31 | 96.60 | 81.39 |
| ENCODER [20] | 46.10 | 77.98 | 87.16 | 94.64 | 76.92 | 90.41 | 95.95 | 77.45 |
| DIPNEC [3] | 47.24 | 80.20 | 89.07 | 97.87 | 73.97 | 89.74 | 95.72 | 77.09 |
| **RUNC (Ours)** | **53.81** | **83.47** | **91.11** | **98.22** | **80.87** | **92.36** | **96.94** | **82.17** |

experiments were implemented in Pytorch on a single NVIDIA A800 GPU and trained for 30 epochs for FashionIQ and 50 epochs for CIRR[2].

## 4.2 Comparison with State of the Arts

Table 1 and Table 2 report quantitative comparisons of our RUNC with the advanced methods on FashionIQ and CIRR datasets, respectively. Detailed architecture descriptions of the compared methods are illustrated in the supplemental material. The proposed RUNC achieves competitive performances on benchmarks of interactive image retrieval. Specifically, a gain of 1.26% on mean R@50 in FashionIQ and a rise of 1.37% on R@10 in CIRR compared with SPRC [31] verify the effectiveness of our proposal. For fashion retrieval, the consistent growth across R@10 and R@50 metrics suggests that this approach accurately grasps user requirements and also adeptly caters to long-tail search demands. Compared with CaLa [22] and SPRC [31] in the same backbone, the overall improvements are significant, as more intrinsic semantic correlations underlying similar candidate fashion images and unclear modification descriptions in fashion queries bring uncertainty when comparing hard negatives and targets. By incorporating evidential distribution to estimate the uncertainty and calibrate the semantic imbalance, RUNC provides more reliable and robust predictions overall. Apart from recall results on all the images in the gallery, the increased metrics on subset settings as 3.95% on Recall$_{subset}$@1 in CIRR compared with the latest ENCODER [20] implicates that our proposed RUNC could identify the authentic intention from the bi-modal query when selecting from a set of closely resembling candidates.

## 4.3 Ablation Analysis

**Analysis of Effective Components.** To assess the efficacy of the model design in this work, ablative results on independent components are shown in Table 3, where "w/o UE" means disabling uncertainty estimation on semantic correlations, "w/o UGC" refers to using original form of contrastive loss without uncertainty coefficients in Eq. 8, "w/o ICM" means removing implicit concept matching, and "w/o SC" means replace the semantic covariance matrix $c$ with ground-truth labels in Eq. 5. The remarkable drop in recall rates of "w/o UE" demonstrates that uncertainty quantification is crucial to boost the performance by perceiving the underlying correlations in candidates, and results of "w/o UGC" further verifies that uncertainty-aware calibration could avoid over-reliance on dominant categories and promote effective learning from negative samples. As the decline of results in "w/o ICM" shows, implicit query guides the distillation of inherent visual elements from reference images by aligning with the targets and independent constraints with textual embeddings. Comparing the proposed model and "w/o SC", the discrepancy highlights the significance of underlying associations in visual candidates and the effectiveness of combining evidential regression with semantic covariance in quantifying semantic ambiguity.

---

[2]Code is available at: RUNC-source.

Table 3: Ablative study on effective components of RUNC.

| Models | FashionIQ | | CIRR | |
|---|---|---|---|---|
| | R@10 | R@50 | R@5 | $R_{subset}$@1 |
| w/o UE | 53.07 | 74.77 | 81.51 | 78.23 |
| w/o UGC | 53.79 | 75.03 | 82.28 | 78.07 |
| w/o ICM | 54.76 | 75.22 | 82.47 | 79.15 |
| w/o SC | 54.19 | 75.14 | 82.30 | 78.88 |
| **Ours** | **55.52** | **76.23** | **83.47** | **80.87** |

Table 4: Analysis of uncertainty estimation on R@50 metric.

| Models | Dress | Shirt | Toptee | Mean |
|---|---|---|---|---|
| GMM | 71.69 | 73.26 | 77.41 | 74.12 |
| BMM | 71.83 | 73.99 | 77.91 | 74.58 |
| EDC | 72.83 | 75.22 | 78.74 | 75.60 |
| MGUR | 72.19 | 74.09 | 78.73 | 75.00 |
| MPC | 72.29 | 74.53 | 78.84 | 75.22 |
| **Ours** | **73.53** | **75.32** | **79.86** | **76.23** |

**Analysis of Uncertainty Estimation.** As shown in Table 4, we also investigate different models to quantify the uncertainty and lead to the following observations. i) "GMM" and "BMM" are sensitive to noise and fail to model aleatoric uncertainty, resulting in unstable retrieval results. Besides, the estimation dependent on the EM algorithm brings computation cost and retrieval latency. ii) Compared to Evidential Dirichlet Classification (EDC) [41], the improvement of RUNC implies that NIG distribution flexibly handles heteroscedastic noise, which better aligns with the coupling semantic correlations in this retrieval task. iii) Though MGUR [4] and MPC [58] utilize probabilistic distribution to model the data uncertainty, insufficient semantic supervision may result in additional noise by nondirective distributions. In comparison, high-order priors in RUNC is superior in quantifying the underlying uncertainty and enhance the interpretability of ranking results.

## 4.4 Further Discussion

**Impact of Uncertainty Supervision.** To evaluate the sensitivity of $\lambda_2$ controlling the uncertainty estimation loss in different datasets, Fig. 3(a) presents recall rates on different settings. The optimal value of $\lambda_2$ for FashionIQ is slightly bigger than CIRR for more severe aleatoric uncertainty issues in FashionIQ. More sensitivity analysis could be referred to in the supplementary material.

**Impact of Evidential Learning.** To verify the necessity of introducing evidential learning to enhance the robustness, we also conduct an ablative setting as directly using semantic covariance matrix $c$ as weights in contrastive learning, denoted as "w/o edl" in Fig. 3(b). Although correlation weights could promote discriminant learning compared with baseline, it is less superior than proposed evidential learning for accumulating errors from precomputed similarities and neglecting the aleatoric uncertainty by data noise and false negatives.

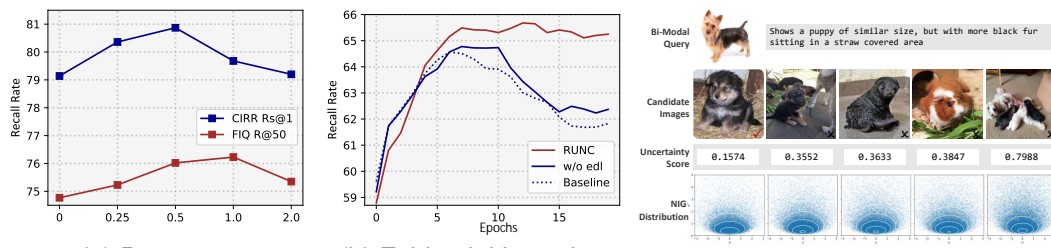

(a) Param $\lambda_2$     (b) Evidential Learning

Figure 3: Analysis of $\lambda_2$ and evidential learning.

Figure 4: Visualization of uncertainty on CIRR.

## 4.5 Visualization Analysis

**Visualization of Uncertainty.** To enhance comprehension of the retrieval results based on uncertainty, we also present retrieval examples to visualize the uncertainty quantification in Fig. 4. For the right-most image with low resolution and sharing semantics like "black fur" and "puppy" with the user intent, the estimated uncertainty value leads to an increase in the penalty by Eq. 8.

**Visualization of Implicit Concept Learning.** Additionally, we also present activation heatmaps by GradCAM [59] on the last module in the vision encoder with other top-ranking results in Figure 5. The prompt embedding $p$ highlights the area corresponding to the dog which is coherent with the reference images, and $m$ underlines the area of the yellow pillow on the couch. The collaboration between $p$ and $m$ enables the model to accurately select the ground-

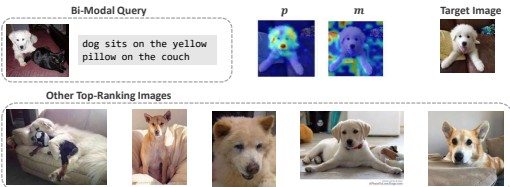

Figure 5: Heatmaps on Target Images with Other Top ranking results.

truth image from other candidates, especially enhancing the perception of object contours, categories, or backgrounds provided by the reference image in $p$.

**Failure Cases.** Failure cases of the proposed RUNC are shown in Figure 6, where the ground-truth images (red boxes) are not ranked the top by our model. For the first example, since the query text in the first instance does not specify the need to change the type of animal in the image, the top-1 image in this model depicts a monkey sleeping on the bend of the tree trunk, which can be regarded as a false negative sample. It also indicates that the proposed RUNC is capable of perceiving the basic query requirements of users effectively. Regarding the second example, due to the complexity of modifying the semantics in the text, and being influenced by factors such as lighting and angles, the model fails to precisely grasp the requirements of "hind legs" and "industrial setting" during measuring the distance between the query and candidate images.

## 5   Conclusion

In this paper, we introduced a novel robust uncertainty calibration model dubbed RUNC to quantify the uncertainty and mitigate the imbalanced semantic learning for interactive image retrieval. To encourage learning from infrequent semantic concepts and mitigate interference caused by semantic overlaps, high-order evidential priors are deployed to estimate the aleatoric and epistemic uncertainty, and target distribution is further adjusted based on uncertainty coefficients. Moreover, we employed an orthogonal loss between explicit textual embeddings and implicit queries to minimize the semantic ambiguity of fusion query from reference images and modification texts. The effectiveness and robustness of RUNC have been demonstrated by extensive experimental results and ablation studies. The potential for extending this approach to other multi-modal learning tasks provides promising prospects for further investigation.

**Limitations.** Since the proposed RUNC utilizes evidential learning that relies on the hypothesis of prior distribution, it may face challenges in continuous learning and lack adaptability to dynamic environments. When the data distribution rapidly changes, frequent updates of uncertain distribution parameters are required, making it difficult for the

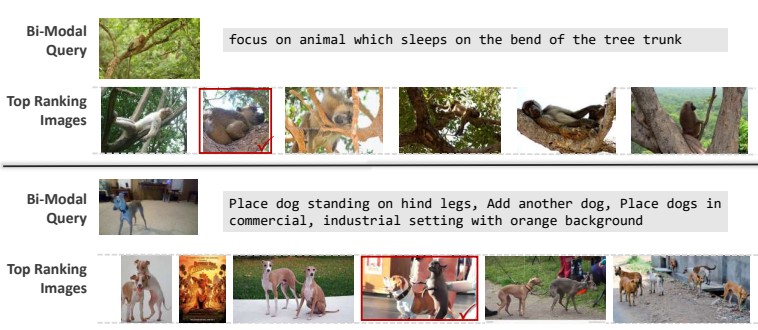

Figure 6: Failure cases in the proposed RUNC.

model to quickly adjust confidence estimates. Besides, a certain amount of data is required in RUNC to fit the prior distribution. In situations involving small sample sizes, the model may struggle to accurately learn the parameters of uncertainty, potentially resulting in either overestimation or underestimation of confidence levels.

## 6   Acknowledgments

This work is supported by the National Key R&D Program of China (2022YFB4701400/4701402), SSTIC Grant (KJZD20230923115106012, KJZD20230923114916032, GJHZ20240218113604008).

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
