# OpenReview forum: "Towards Robust Uncertainty Calibration for Composed Image Retrieval"
_NeurIPS.cc/2025/Conference — NeurIPS 2025 poster_

### Official Review · Reviewer_gEs2 · 2025-06-27

**Clarity:** 2
**Significance:** 2
**Originality:** 3
**Rating:** 5
**Confidence:** 3

**Summary:**

This paper proposes to adopt uncertainty estimation into Composed Image Retrieval problems to address the problem of ambiguous and imbalanced semantics of the bi-modal queries. High-order evidential priors are employed for semantic distribution adjustment. Orthogonal independent constraints are applied to two branches of queries for disambiguation. Experiments on common datasets show its effectiveness under multiple metrics.

**Questions:**

1. Why is it necessary to adopt an uncertainty estimation method for the problem?
2. Why does the learnable prompts have unclear semantic concepts (Sec. 3.3)? Is there any evidence?
3. What are the differences between the two branches of frozen and trainable text encoders? What are the differences between Eq.(1) and Eq.(11)? The variables are matched and using one single branch may also work.

**Ethical Concerns:**

["NO or VERY MINOR ethics concerns only"]

**Final Justification:**

The rebuttal clarified the motivation of key designs in their work, provided experiments to justify the paper's declarations, and improved on many technical issues. Therefore, I decide to raise my score to accept.

**Limitations:**

yes

**Paper Formatting Concerns:**

No major formatting issues

**Quality:**

3

**Strengths And Weaknesses:**

Strength:
 - The method was rigorously designed and theoretically analyzed to solve the significant uncertainty problem in CIR with great novelty.
 - The Experiments section have compared the proposed method with numerous methods on various metrics, while presenting ablations and discussions on both the proposed components and uncertainty estimation.

Weaknesses:
1. Clarity and motivation:
 - Some parts lack detailed mathematical definitions (assumption at Line 161-162), motivation statements, and citations of the core method used (Section 3.2), making the method section hard to read.
 - Temperature $\tau$ is often writen as divided rather than multiplied due to it's meaning. In the paper it is written as multiplied.
2. There is no subscript for variables in Eq.(12). Meanwhile, it is recommended to explain the meaning and effect of Eq.(12) as it's good for the clarity of the proposed Implicit Concept Matching.

---

> ### Author Rebuttal · Authors · 2025-07-31
>
> ***Q1: Why is it necessary to adopt an uncertainty estimation method for the problem?***
>
> **A1:** Thank you for raising this important problem. We clarify the significance of uncertainty estimation and our design as follows:
> - In the setting of composed image retrieval, candidate images **exhibit underlying semantic overlaps** with the targets as the images inherently contain rich semantic concepts. This phenomenon is more pronounced in fashion datasets, potentially leading to false negatives. The latent correlation between the matched and mismatched instances causes difficulty for the model to discriminate the positives from the negatives. Thus, we estimate the uncertainty scores of instances to enhance the distinction of hard negatives.
> -  The imbalanced distributions of semantic concepts in visual features tend to cause **overfitting and overconfidence in dominant semantics**. Thus, the ranking results are inclined to be misguided by partially related semantics while ignoring discriminant details as the user requires (as seen in Figure 1(left)). The overconfidence in partial characteristics could be quantified by **epistemic uncertainty** based on the evidential learning theory.
> - Vague expression of modification requirements (query texts), low-resolution and totally irrelevant reference images (query image), and noisy labels (some matched images are not annotated as positives) introduce **aleatoric uncertainty (uncertainty in data)** in this task and motivate us to capture the intrinsic semantic alignments while reducing the uncertainty in similarity distributions.
>
> ***Q2: Why do the learnable prompts have unclear semantic concepts (Sec. 3.3)? Is there any evidence?***
>
> **A2:** Thank you for raising this important concern. Please let us further clarify the generation of learnable prompts $\boldsymbol{p}$. Q-Former structure employs the initialized learnable queries to attend to the image regions for aligning with texts. In the setting of composed image retrieval, the modifiers specify changes to reference images. In other words, the input image (reference image) and the input text (modification text) **are not semantically aligned**, and they usually have contradictory semantics. However, 1) the Q-Former structure based on multi-head attention mechanisms attempts to align the query token with the image patches and **forces the alignment between the inconsistent input images and texts**. 2) Since the query token of Q-Former is a trainable embedding that attempts to capture local semantics of the image and further fuse with texts, the composed feature is inclined to **be misleadingly guided and reserves visual regions irrelevant to the targets** when the textual description contradicts the image content. 3) The traditional contrastive loss simply pushes apart mismatched pairs and brings matched pairs closer, which **lacks strong supervision signals** to penalize embeddings that contain incorrect semantics. The above reasons theoretically lead to the unclear semantic concepts, which motivate us to design the implicit concept matching (ICM) module to minimize the ambiguity when fusing queries.
>
> Besides, the comparison of aleatoric uncertainty scores of the complete RUNC in comparison with the model without the ICM module. The model without ICM generally has large uncertainty scores and struggles with training convergence, implying that the learnable prompts inherently have unstable and ambiguous semantic representation, and the proposed model could alleviate the uncertainty issues by incorporating implicit guidance.
>
> Models|Epoch 0|Epoch 5|Epoch 10|Epoch 15|Epoch 20
> ----|----|----|----|----|----
> w/o ICM|0.902|0.614|0.334|0.199|0.209
> w ICM|0.692|0.234|0.118|0.086|0.086
>
> ***Q3: What are the differences between the two branches of frozen and trainable text encoders? What are the differences between Eq.(1) and Eq.(11)?***
>
> **A3:** Thanks for raising this question. We apologize for the confusion regarding the design of the implicit concept matching (ICM) module. In our proposed RUNC, one branch yields visual prompts $\boldsymbol{p}$ and fusion query $\boldsymbol{q}$ for uncertainty-based matching, while the other branch (i.e., ICM) introduces symmetrical embeddings to guide prompts $\boldsymbol{p}$, aiming to reduce uncertainty in $\boldsymbol{q}$. Both branches share the same text encoder architecture, and ICM uses shared weights to ensure feature consistency and avoid noisy gradients. We will revise Figure 2 and notations to improve clarity.
>
> Regarding Eq.1 and Eq.11: Eq.1 defines a standard contrastive loss on matching scores $\boldsymbol{s}$ between fusion query $\boldsymbol{q}$ and target $\boldsymbol{t}$, but it **is not used in our framework**, as it treats all negative samples equally and overlooks latent semantic overlap. Instead, we use Eq.8 to adjust the candidate distribution based on estimated uncertainty, penalizing uncertain instances. In contrast, Eq.11 **imposes constraints on $\boldsymbol{s}^\ast$** to distill visual semantics from the target $\boldsymbol{t}$ to the implicit query $\boldsymbol{q}^\ast$ in the ICM module. This enhances the representation of $\boldsymbol{p}^\ast$, which shares the same shape as prompt $\boldsymbol{p}$, and guides its learning via $\mathcal{L}_{al} = \lVert \boldsymbol{p} - \boldsymbol{p}^\ast \rVert$. Removing the ICM branch results in prompts retaining more noise and irrelevant features. Ablation results in Table 3 confirm that the implicit query branch effectively distills target-aware visual elements and strengthens alignment through independent constraints with text embeddings.
>
> ***W1: Detailed mathematical definitions (assumption at Line 161-162), motivation statements, and citations of the core method used (Section 3.2)***
>
> **A4:** Thank you for your valuable feedback. We are sorry for the confusion in the uncertainty estimation.
>
> Given the initial matching scores between the composed queries and candidate instances, the Uncertainty Estimation module in Section 3.2 aims to maximize the likelihood of our observations. In the context of evidential learning [1,2,3], the latent semantic correlations are drawn i.i.d from distributions with unknown mean and variance parameters. The detailed mathematical definitions could be referred to Eq. 3 as: $\boldsymbol{c}_i \thicksim \mathcal{N}(\mu_i, \delta_i^2)$, where $\boldsymbol{c}$ means the semantic covariance scores.
> The core motivation for this part lies in **semantic overlaps within candidate images**. As Figure 1(left) shows, negative candidate image $\boldsymbol{I}\_4$ shares strong semantic correlations with the targets $\boldsymbol{I}\_ {gt}$ and user requirements, while $\boldsymbol{I}_4$ is annotated as negatives. The matching score between $\boldsymbol{I}_4$ and the query is expected to be 0 in traditional contrastive loss, hindering the semantic distance measurement in the latent space. Instead, we employ the semantic covariances $\boldsymbol{c}$ as a soft supervision to guide the distributions of match scores, and quantify the uncertainty scores for candidates from evidential learning. We would reorganize the beginning of Section 3.2 to improve the readability of the method and add corresponding works in the illustrations.
>
> ***W2: Temperature is often written as divided rather than multiplied.***
>
> **A5:** Thanks for your advice. We are sorry for the lack of standardization in the formulas and have revised the corresponding formulas as:
> $\mathcal{L}\_{cl}=-\frac{1}{B} \sum\_{i=1}^{B} \log \frac{\exp(\boldsymbol{s}\_{ii} /\ \tau)}{\sum\_{j=1}^{B} \exp(\boldsymbol{s}\_{ij}/\\tau)}$ (1), $\mathcal{L}\_{bc}=-\frac{1}{B} \sum\_{i=1}^{B} u_i \cdot \log \frac{\exp(\boldsymbol{s}\_{ii} /\ \tau)}{\sum\_{j=1}^{B} \exp(\boldsymbol{s}\_{ij}) /\ \tau }$ (8), and $\mathcal{L}\_{ce}=-\frac{1}{B} \sum\_{i=1}^{B} \log \frac{\exp(\boldsymbol{s}\_{ii}^* /\ \tau)}{\sum\_{j=1}^{B} \exp(\boldsymbol{s}^*\_{ij}/\\tau)}$ (11).
>
> These formulas would be updated in the revised version.
>
> ***W3: Explanation of Eq.12***
>
> **A6:** Thanks for your questions, and we are sorry for the confusion in Eq.12. The revised formula for Eq.12 is：
> $\mathcal{L}\_{ort}=\sum\_{i\neq j} \left(\text{Cov}(\boldsymbol{p}^\ast, \boldsymbol{m})\_{ij}\right)^2=\sum\_{i \neq j} \left( \frac{1}{B} ((\boldsymbol{p}^\ast)^\top \boldsymbol{m} \right)\_{ij})^2$.
>
> The orthogonal loss in Eq.12 aims to penalize the correlations between features $\boldsymbol{p}^\ast$ and $\boldsymbol{m}$ in different dimensions. In CIR task, $\boldsymbol{m}$ from text tokens usually have explicit modification requirements, while the reference images are coupled with the visual semantics to be retained, removed, and any redundant information. The constraints in Eq.12 avoid feature redundancy and **compel the implicit query $\boldsymbol{p}^\ast $ to learn complementary information and prevent feature overlap with text**. Hence, the implicit query $\boldsymbol{p}^\ast$ could further guide the actual visual prompts $\boldsymbol{p}$ to distill effective retention information from reference images. We would add more illustrations of Eq.12 in the revised version.
>
>
> Refs:
>
> [1] Malinin et al. Predictive Uncertainty Estimation via Prior Networks. NeurIPS 2018: 7047-7058
>
> [2] Sensoy et al. Evidential Deep Learning to Quantify Classification Uncertainty. NeurIPS 2018: 3183-3193
>
> [3] Amini et al. Deep Evidential Regression. NeurIPS 2020
>
> We generally thank you for your time and effort in reviewing our manuscript. Thank you again for your valuable feedback.

---

> > ### Comment · Reviewer_gEs2 · 2025-08-06
> > **Response**
> >
> > Thanks for the response. The author rebuttal has addressed most of my concerns and I appreciate their detailed responses. In light of these clarifications and revisions, I am raising my score accordingly.

---

> > > ### Author Response · Authors · 2025-08-06
> > > **Response to Reviewer gEs2's Supportive Feedback**
> > >
> > > Thanks to Reviewer gEs2 for the positive feedback and for raising the score of our manuscript. We will continue to refine the manuscript and incorporate your valuable suggestions into our revised paper.

---

### Official Review · Reviewer_gQCa · 2025-06-29

**Clarity:** 3
**Significance:** 2
**Originality:** 3
**Rating:** 4
**Confidence:** 4

**Summary:**

This paper addresses the challenge of semantic ambiguity and uncertainty in composed image retrieval (CIR), where a retrieval query consists of a reference image and a modification text. The authors identify that prior contrastive learning-based methods often fail to ensure reliable matching due to both aleatoric and epistemic uncertainty arising from implicit semantic overlaps and dataset biases. To tackle this, they propose RUNC (Robust UNcertainty Calibration), a novel framework that estimates and calibrates uncertainty using evidential priors, such as Normal-Inverse Gamma distributions, to balance semantic alignment. RUNC also introduces orthogonal constraints between explicit textual embeddings and implicit query signals to reduce redundancy and ambiguity in the fused query representation. Experiments on benchmark datasets like FashionIQ and CIRR demonstrate that RUNC achieves more robust and reliable retrieval rankings.

**Questions:**

1.Is the Gaussian and i.i.d. assumption on semantic matching scores reasonable? The method assumes that semantic similarities follow an independent Gaussian distribution, which may not hold in real-world multimodal data with contextual and correlated semantics. Have the authors tested the robustness of the model when this assumption is relaxed?
2.How essential is Q-Former to the overall performance? The Q-Former is reused from BLIP-2 with limited adaptation. Could the authors provide a baseline using simpler fusion (e.g., concatenation or attention) to clarify its actual impact?
3.Are the learnable prompts in Q-Former interpretable? Given the role of learnable prompts in query construction, could the authors provide attention maps or qualitative analyses to show what these prompts are attending to?
4.Could a progressive ablation be added to show module synergy? Instead of only showing removal of modules, a progressive build-up of the model (e.g., Base → +UE → +UGC → Full) would help demonstrate how each component contributes incrementally.

**Ethical Concerns:**

["NO or VERY MINOR ethics concerns only"]

**Final Justification:**

After carefully reading the authors' response, I found that for Q1, the authors addressed my confusion regarding the independent and identically distributed Gaussian distribution assumption. Their model does not assume global independence, but instead uses a normal-inverse gamma (NIG) distribution to model the uncertainty of each query, which aligns well with multimodal contextual dependencies. For Q2, the authors provided compelling experiments demonstrating the role and necessity of the Q-Former. For Q3, the authors added qualitative analysis in the supplementary materials, enhancing the interpretability of the learnable cues. For Q4, the authors demonstrated the cumulative benefits of their proposed module through comprehensive ablation experiments, validating the effectiveness of their approach. Overall, the authors provided a clear response, addressing key concerns, resolving my concerns, and improving the persuasiveness of their article.

**Limitations:**

Yes

**Paper Formatting Concerns:**

No major formatting issues identified. The paper follows NeurIPS guidelines appropriately.

**Quality:**

3

**Strengths And Weaknesses:**

This paper proposes a novel uncertainty-aware composed image retrieval framework that integrates Gaussian regression modeling with implicit query supervision. The method demonstrates solid technical quality, with well-structured theoretical derivations and comprehensive ablation studies. The writing is generally clear, though the explanation of some component abbreviations (e.g., UE, UGC) could be improved.  The proposed approach addresses practical challenges in multi-modal retrieval, such as semantic ambiguity and label imbalance, and shows competitive performance on two benchmarks. The use of Normal Inverse Gamma distribution for uncertainty modeling and virtual guidance for implicit query alignment are relatively novel, although further clarification of the difference from prior contrastive methods would enhance the contribution. Overall, the paper is technically sound and presents meaningful and original contributions to the field.

---

> ### Author Rebuttal · Authors · 2025-07-31
>
> ***Q1: Is the Gaussian and i.i.d. assumption on semantic matching scores reasonable? The method assumes that semantic similarities follow an independent Gaussian distribution, which may not hold in real-world multimodal data with contextual and correlated semantics.***
>
> **A1:** Thank you for this insightful question. We agree that semantic similarities in multimodal data may exhibit contextual correlations. This is also why we incorporate semantic covariance $\boldsymbol{c}$ to measure the correlation of semantic concepts. However, our proposed model **does not assume global i.i.d. Gaussian distributions across all semantic pairs**. Rather, we follow the evidential modeling paradigm [1,2] using the Normal-Inverse-Gamma (NIG) distribution to **capture uncertainty in semantic matching for each query sample**. Specifically, for the $i$-th instance, we consider the semantic correlations with all candidate samples as a random variable  $\boldsymbol{c}_i$ and is sampled from a Gaussian distribution $\boldsymbol{c}_i \thicksim \mathcal{N}(\mu_i, \delta_i^2).$ We further model the uncertainty in both $\mu_i$ and $\delta_i^2$ via a Normal-Inverse-Gamma prior. This assumption does not require independence across different samples, but rather treats the matching distribution for each query as conditionally independent.
>
> Fitting semantic features in multimodal data via high-dimensional Gaussian models has been **widely adopted in various models** in fields such as vision, language, and image-text alignment [3,4,5]. In this work, we incorporate NIG distribution to quantify the uncertainty in semantic alignment, where the mean could be regarded as the semantic center, and the covariance and statistical moments in Eq.6 characterize uncertainty. Using distribution matching instead of point vectors could help alleviate overfitting issues and improve the robustness of the model. Besides, high-order conjugate prior for the Gaussian distribution could **effectively capture the "one-to-many" semantic correlations**, which is consistent with the nature of the multi-modal retrieval process.
>
> ***Q2: How essential is Q-Former to the overall performance? Could the authors provide a baseline using simpler fusion (e.g., concatenation or attention) to clarify its actual impact?.***
>
> **A2:** Thank you for your questions. Q-Former in our proposed RUNC serves as a fusion module to extract aligned visual-textual semantics from the reference image and modification sentences. We would like to clarify that the **Q-Former is not the core contribution and novelty of our method**. Rather, we employ Q-Former in this work primarily for its strong capability to **extract semantically rich visual embeddings** from pre-trained frozen vision encoders, which facilitate semantic alignments to mitigate the heterogeneous gap. Besides, the employment of the Q-Former structure is in line with advanced models such as CaLa[6] and SPRC[7]. Our contribution concentrates on uncertainty-guided semantic alignment and calibration on the similarity distributions built on top of the multi-modal query fusion.
>
> To evaluate the impact of Q-Former, we have conducted additional experiments replacing the Q-Former with a simpler fusion mechanism, as shown in the Table below. “Concat” means concatenation of image and text features followed by a shared MLP. It can be seen that simple fusion strategies underperform our proposed approach in both datasets, as they have difficulty in fully capturing and integrating complex visual and linguistic features. Nonetheless, our uncertainty-aware mechanism does not depend on the Q-Former structure and is compatible with alternative fusion strategies.
>
> Models | FIQ R@10 | FIQ R@50 | FIQ Avg. | CIRR R@5 | CIRR $\mathbf{R}_{subset}@1$ | CIRR Avg.
>  ----  | ----  |----  |---- |---- |---- |----
> Concat  | 47.67 | 69.75 | 58.71 | 80.15 | 76.16 | 78.16
> Single Attention Layer | 48.97  | 71.63 | 60.30 | 80.94 | 77.03 | 78.99
> Q-Former  | 55.52 | 76.23 | 65.88| 83.47 | 80.87 | 82.17
>
> ***Q3: Are the learnable prompts in Q-Former interpretable? Given the role of learnable prompts in query construction, could the authors provide attention maps or qualitative analyses to show what these prompts are attending to?***
>
> **A3:** Thank you for raising this important concern. In our implementation, the Q-Former takes the input of image embeddings of the reference image from a frozen ViT encoder, text tokens of modification sentences, and a fixed set of learnable query tokens. The learnable prompts $\boldsymbol{p}$, as the outputs of Q-Former, encode attentive visual images by utilizing the initialized learnable queries to attend to the image regions for aligning with texts. For this specific task, the learnable prompts $\boldsymbol{p}$ target at **enhancing visual representations of the semantics preserved in the reference images** (e.g., basic outlines of the fashion items, main objects and species, the reversed attributes).
>
> Attention maps and qualitative analysis are essential for providing a vivid presentation of learnable prompts. Due to the length limit of the main manuscript, we presented **activation heatmaps by GradCAM in Figure 8 in the supplementary materials**. As shown in the first example of Figure 8, the learnable prompts $\boldsymbol{p}$ highlight the area corresponding to the dog, which is coherent with the reference images. In contrast, the text features $\boldsymbol{m}$ underlines the area of the yellow pillow on the couch. The collaboration between $\boldsymbol{p}$ and $\boldsymbol{m}$ enables the model to accurately select the ground-truth image from other candidates, especially enhancing the perception of object contours, categories, or backgrounds provided by the reference image in $\boldsymbol{p}$.
>
> ***Q4: Could a progressive ablation be added to show module synergy?***
>
> **A4:** Thank you for your suggestions. To address your concern, we have conducted ablative experiments by incrementally adding modules to the base model in the table below. “UE” refers to adding the constraints of $\mathcal{L}^{NLL}$ proposed in the uncertainty estimation. “UGC” refers to assigning uncertainty coefficients to calibrate the distributions based on the quantified uncertainty scores. “ICM” refers to adding the Implicit Concept Matching module to guide the reference images. Note that the Uncertainty-Guided Calibration(UGC) is not an independent module, and instead is built on the Uncertainty Estimation(UE) module to penalize uncertainty samples adaptively on the estimated uncertainty scores.
>
> The qualitative results of adding UGC bring the most remarkable results when comparing other individual modules, suggesting the **significance of considering uncertainty in match scores** and emphasizing contrastive learning from uncertain instances. The UE module introduces the NIG distribution to fit the semantic covariances, which is the basis of distinguishing uncertain samples. Though it does not directly refine the similarity distributions, it still brings improvements especially in fashion dataset, as the semantic correlations serve as soft supervisions to promote latent alignment learning. The growth of ICM also implies that implicit concept matching could mitigate the ambiguity of fusion queries. The integration of all the modules aims to alleviate uncertainty issues in query compositions and query-to-target matching.
>
> Models | FIQ R@10 | FIQ R@50 | FIQ Avg. | CIRR R@5 | CIRR $\mathbf{R}_{subset}@1$ | CIRR Avg.
>  ----  | ----  |----  |---- |---- |---- |----
> Baseline  | 51.45  |  73.09  | 62.27 |  79.08  |  75.48  | 77.28
> +UE  |  53.28  | 75.11 |  64.20  |  79.79  |  77.52  |  78.66
> +UGC  |  54.76  | 75.22 |  64.99  |  82.47  |  79.15  |  80.81
> +ICM  |  53.07  | 74.77 |  63.92  |  81.51  |  78.23  | 79.87
> Ours(all)  | 55.52 | 76.23 | 65.88| 83.47 | 80.87 | 82.17
>
> Refs:
>
> [1] Sensoy et al. Evidential Deep Learning to Quantify Classification Uncertainty. NeurIPS 2018: 3183-3193
>
> [2] Amini et al. Deep Evidential Regression. NeurIPS 2020
>
> [3] Kendall et al. What Uncertainties Do We Need in Bayesian Deep Learning for Computer Vision? NIPS 2017: 5574-5584
>
> [4] Chun et al. Probabilistic Embeddings for Cross-Modal Retrieval. CVPR 2021: 8415-8424
>
> [5] Huang et al. Cross-Modal Recipe Retrieval With Fine-Grained Prompting Alignment and Evidential Semantic Consistency. IEEE Trans. Multim. 27: 2783-2794
>
> [6] Jiang et al. CaLa: Complementary Association Learning for Augmenting Comoposed Image Retrieval. SIGIR 2024: 2177-2187
>
> [7] Bai et al. Sentence-level Prompts Benefit Composed Image Retrieval. ICLR 2024
>
> We sincerely appreciate your review of our manuscript

---

> ### Comment · Reviewer_gQCa · 2025-08-06
> **Response**
>
> Thank you for your response. After carefully reviewing the authors' response and the supplementary experiments provided, my confusion has been resolved. After reassessing the paper, I am now confident that it meets the conference's review criteria and will update my review rating accordingly.

---

> > ### Author Response · Authors · 2025-08-06
> > **Response to Reviewer gQCa's Feedback**
> >
> > Thanks to Reviewer gQCa for the valuable feedback on our manuscript. We appreciate the time and effort you have dedicated to reassessing our submission. We will continue to refine the manuscript and incorporate your constructive feedback into the final version.

---

### Official Review · Reviewer_K9V5 · 2025-06-30

**Clarity:** 2
**Significance:** 3
**Originality:** 4
**Rating:** 4
**Confidence:** 4

**Summary:**

A novel method named RUNC (Robust Uncertainty Calibration) is proposed in this paper. The core idea of RUNC is to achieve more robust semantic matching by quantifying and calibrating uncertainty. This approach specifically addresses the unreliable matching results found in existing methods for Composed Image Retrieval tasks, a limitation primarily stemming from aleatoric and epistemic uncertainty. Aleatoric uncertainty stems from the data itself, while epistemic uncertainty is typically caused by overconfidence in dominant semantic categories. RUNC introduces Uncertainty Quantification and Semantic Calibration, alongside an Implicit Concept Matching module, to address these challenges.

**Questions:**

1. The specific network structure of the "Uncertainty Perceptron" module, including its number of layers, dimensions, and activation functions, is not sufficiently detailed in the main text. Could the authors provide a more precise architectural description of this component?
2. Given that the paper's core is "Robust Uncertainty Calibration," how robust is RUNC to input noise (e.g., slight corruption in reference images, typos in modification texts) or even adversarial attacks, beyond handling inherent data uncertainty? Does the model's uncertainty estimation remain accurate under these conditions?
If the authors can provide convincing experimental results demonstrating RUNC's robustness to input noise and adversarial attacks and accuracy of uncertainty estimation under these conditions, my Significance scores will increase.
3. Considering that RUNC introduces additional modules, could the authors provide a more detailed analysis of its computational efficiency, including comparisons of training and inference times against key baselines? This would strengthen the argument for its practical applicability.
4. The explanation of the implicit query's generation process, "virtual guidance," could be more concrete. Specifically, could the authors clarify whether this virtual guidance is a learnable parameter, how it is initialized, and what mechanisms ensure its effectiveness in distilling visual semantics without redundant textual information? A pseudo-code or more detailed algorithmic steps would be highly beneficial.
If the authors can provide a clear and precise architectural description for the "Uncertainty Perceptron" module and offer a concrete explanation of the implicit query's generation process and mechanisms, my Clarity score will increase.

**Ethical Concerns:**

["NO or VERY MINOR ethics concerns only"]

**Final Justification:**

After careful consideration of their responses and alignment with fellow reviewers, I will maintain my original rating.

**Limitations:**

yes

**Quality:**

3

**Strengths And Weaknesses:**

Strengths：
1. The RUNC framework, based on high-order Normal Inverse Gamma distributions for quantifying aleatoric uncertainty and epistemic uncertainty, is built upon a solid theoretical foundation.
2. The ablation studies are very thorough, clearly demonstrating the positive contributions of RUNC's various components: Uncertainty Estimation, Uncertainty-Guided Semantic Calibration, Implicit Concept Matching, and Semantic Covariance, to the overall performance.
3. The mathematical principles and workflow of RUNC's various modules – Uncertainty Perceptron, Uncertainty-Guided Semantic Calibration, Penalty on Misleading Evidence, and Implicit Concept Matching – are described in detail, complemented by an intuitive framework diagram (Figure 2).
4. The paper innovatively introduces the NIG distribution, utilized for uncertainty quantification in evidential deep learning, into the specific task of CIR, skillfully integrating it with CIR task characteristics for handling semantic covariances and imbalanced semantics.

Weaknesses:
1. The theoretical analysis of "high-order conjugate prior distributions" could be more rigorously elaborated in the main text or provided as a more comprehensive overview in the supplementary material.
2. A more comprehensive comparison with some of the latest SOTA methods is needed to more fully demonstrate the competitiveness of the proposed method.
3. The specific network structure of the "Uncertainty Perceptron" module, including its number of layers, dimensions, and activation functions, is not sufficiently detailed in the main text. The explanation of the implicit query's generation process could be more concrete, including whether it is learnable, how it is initialized, and how its effectiveness in distilling visual semantics without redundant textual information is ensured.

---

> ### Author Rebuttal · Authors · 2025-07-31
>
> ***Q1/W3: Architectural description of uncertainty perception.***
>
> **A1:** Thank you for this valuable suggestion. The uncertainty perceptron module is implemented with two linear layers followed by a Softplus activation function.  The dropout rate is set as 0.2. The input and hidden dimensions, denoted as $N_q$, are set equal to the number of learnable queries, fixed at 32 in our implementation. The output dimension is $4N_q$, which is split to predict the NIG parameters $\gamma$, $\nu$, $\alpha$, and $\beta$. With the guidance of semantic covariances of candidates, the inferred probabilistic distributions from the uncertainty perceptron estimate aleatoric and epistemic uncertainty in each candidate and emphasize learning on uncertain negative images. We would add a detailed implementation of the uncertainty perceptron in the revised version.
>
> ***Q2: How robust is RUNC to input noise.***
>
> **A2:** Thanks for raising this important question. We would address your concerns as follows:
> - Our work focuses on quantifying semantic uncertainty in composed image retrieval using a Normal-Inverse-Gamma (NIG) distribution to model both aleatoric and epistemic uncertainty. **Aleatoric uncertainty** arises from vague modification expressions, low-resolution references, or noisy annotations, while **epistemic uncertainty** reflects overconfidence in dominant semantics. When input noise (e.g., image corruption or typos) disrupts semantic alignment, our model increases predicted uncertainty, avoiding overfitting to corrupted inputs.
> - To further investigate the robustness of our proposed model, we have conducted additional experiments in FashionIQ under two categories of noise: (1) Visual noise (V): introduce Gaussian noise to obtain the jittered visual representations. (2) Textual noise (T): randomly swap the characters of words in modifications. The compared results in two noise settings reveal that the model demonstrates greater sensitivity to textual noise, which significantly disrupts the users' core requirements, particularly in attribute modifications. As for the visual noise, the reference image inherently contains contradictory and redundant semantics. With our designed uncertainty-guided calibration and implicit concept alignment, the proposed model could quantify the uncertain semantic alignments and utilize reliable alignment via implicit matching to compose robust features. Generally, these results support the robustness of our framework even in the presence of externally introduced noise.
>
> Models|Dress|Shirt|Toptee|Avg R@50
> ----|----|----|----|----
> Baseline(V)|72.83|71.77|74.34|72.98
> Ours(V)|73.14|73.36|77.24|74.58
> Baseline(T)|70.11|70.09|75.44|71.88
> Ours(T)|71.44|72.57|77.77|73.93
>
> ***Q3: More analysis of computational efficiency, including comparisons of training and inference times.***
>
> **A3:** Thank you for the suggestion. To address the concern of computation costs, we compare the training time and inference time in the table below. The comparison results are tested in a single A800 GPU on FashionIQ, and the “training time” means the average training time for all the triplets in one epoch. Specifically, CaLa[1] incorporated twin visual compositors and constructed complementary associations, which introduces computation latency both in training and inference time. Compared with SPRC[2], our proposed model brings limited computation complexity during training, since we only introduce a light-weight uncertainty-based perceptron and the implicit matching shares similar structures with the query fusion. As for the inference stage, the candidate features are indexed from the frozen visual encoders in advance to save inference time. Thus, the implicit concept matching is disabled, and the matching scores can be directly calculated from the cosine distance between the query fusion feature and the candidate features. Generally, our proposed model **exhibits computational efficiency comparable to state-of-the-art methods**.
>
> Models|Training Time(s)|Inference Time(s)
> ----|----|----
> CaLa[1]|263.1|45.7
> SPRC[2]|209.5|27.2
> Ours|214.3|27.4
>
> ***Q4/W3: The explanation of the implicit query generation process.***
>
> **A4:** Thank you for this insightful comment.  The virtual guidance $\boldsymbol{p}^\ast$  is initialized with a zero-mean Gaussian distribution with a standard deviation specified by the Q-Former configuration, and it is updated during end-to-end training. The detailed algorithm steps are:
>
> 1.Initialize learnable embedding $\boldsymbol{p}^\ast$ with Gaussian distribution.
>
> 2.Expand $\boldsymbol{p}^\ast$ to batch size.  # keeps the same shape size as $\boldsymbol{p}$.
>
> 3.Generate implicit query $\boldsymbol{q}^{*}$： use shared text encoders to integrate modification text tokens and learnable virtual guidance $\boldsymbol{p}^\ast$ and normalize the output $\boldsymbol{q}^\ast$.
>
> 4.Compute cosine similarity scores $\boldsymbol{s}^\ast$ based on implicit fusion query $\boldsymbol{q}^\ast$ and target embeddings $\boldsymbol{t}$.
>
> 5.Compute contrastive loss $\mathcal{L}\_{ce}$, alignment loss $\mathcal{L}\_{al}$, and orthogonal loss $\mathcal{L}\_{ort}$.
>
> The effectiveness in distilling informative visual semantics is ensured by the joint cooperation of constraints and symmetrical design to provide implicit guidance. Specifically, $\mathcal{L}\_{ce}$ align targets with the implicit query $\boldsymbol{q}^\ast$. Through the shared encoders and similar inputs, the learnable virtual guidance $\boldsymbol{p}^{*}$ is expected to maintain the semantics consistent with the targets. Then the orthogonal loss $\mathcal{L}\_{ort}$ forces virtual guidance $\boldsymbol{p}^\ast$ to be independent with text embeddings $\boldsymbol{m}$, to **remove overlapping semantics with the text in $\boldsymbol{p}^\ast$  and preserve maximum information from the reference images**. $\mathcal{L}\_{al}$ further aligns virtual guidance $\boldsymbol{p}^\ast$ with the original visual prompts $\boldsymbol{p}$. The above process guarantees the distillation of visual semantics without redundant textual information.
>
> Besides, ablation experiments in Table 3 and visualization of implicit matching in Figure 6,7, and 8 also demonstrate that this design could reduce redundant information in the prompt, ensuring efficient extraction of information that matches the target.
>
> ***W1: Theoretical analysis of high-order conjugate priors needs more rigor.***
>
> **A5:** Thank you for your suggestions. The conjugate prior is defined as: if prior $p(\theta)$ and likelihood $p(x \mid \theta)$ form a conjugate pair, the posterior $p(\theta \mid x)$ remains in the same family. High-order conjugate priors extend this consistency to complex likelihoods. The NIG is a second-order conjugate prior modeling unknown mean $\mu$ and variance $\delta^2$.
>
> Given data $\boldsymbol{x} \sim \mathcal{N}(\mu, \delta^2)$, the likelihood is: $p(\boldsymbol{x}|\mu, \delta^2) = \frac{1}{\sqrt{2\pi\delta^2}}\exp \left(-\frac{(x-\mu)^2}{2\delta^2}\right)$.
> This can be rewritten as an exponential family form: $p(x|\boldsymbol{\theta}) \propto \exp\left( \begin{bmatrix} \frac{\mu}{\sigma^2} \ -\frac{1}{2\sigma^2} \end{bmatrix}^\top \begin{bmatrix} x \ x^2 \end{bmatrix} + \Phi(\boldsymbol{\theta}) \right)$, with $\boldsymbol{\theta}=(\mu, \delta^2)$ and sufficient statistics $\mathbf{T}_x=(x, x^2)$.
> The prior (as in Eq. 4) is: $p(\mu, \delta^2) \propto (\delta^2)^{-(\alpha + \frac{1}{2}) - 1} \exp\left(\begin{bmatrix} \frac{\nu\gamma}{\sigma^2} \ -\frac{\nu}{2\delta^2} \end{bmatrix}^\top \begin{bmatrix} \mu \ \mu^2 \end{bmatrix} - \frac{1}{\delta^2} \left( \beta + \frac{\nu\gamma^2}{2} \right)\right)$.
>
> As the kernel of the NIG distribution matches the exponential family form of the Gaussian likelihood, the posterior distribution remains NIG. Besides, the updates of its hyperparameters rely on second-order statistics. The **detailed updation of parameters of $\gamma,\nu, \alpha, \beta $ could be referred to Section A.3 in the supplementary details**.
>
> In our proposed RUNC, the NIG prior models full posteriors over mean and variance, providing richer uncertainty than point or diagonal-Gaussian models. This hierarchical prior captures both aleatoric and epistemic uncertainty (Eq. 6, 7) and reflects prediction confidence and ambiguity spread. We will provide more theoretical analysis in the final version.
>
> ***W2: More comprehensive comparison with some of the latest SOTA methods.***
>
> **A6:** Thank you for your suggestions. We have added overall results comparison with CoPE [3] and CCIN [4] in the table below. Specifically, CoPE used Gaussian distributions to represent queries and targets. CCIN utilized LLM-based analysis to identify conflicting attributes and generate instructions.  Our RUNC still demonstrates competitive performance on two datasets. Though R@5 and R@10 scores of CCIN[5] are slightly higher, CCIN introduced a Large Language Model to facilitate analysis on conflict semantics and performed LoRA fine-tuning on the ViT backbone. In comparison, our proposed model is a concise end-to-end training framework dedicated to distribution calibration based on uncertainty.
>
> Methods|FIQ R@10|FIQ R@50|FIQ Avg.|CIRR R@1|CIRR R@5|CIRR R@10|CIRR R@50|CIRR Avg.
> ----|----|----|----|----|----|----|----|----
> CoPE[1]|44.50|68.60|56.55|49.18|80.65|89.86|98.05|79.55
> CCIN[2]|54.41|74.76|64.59|53.41|84.05|91.17|98.00|81.66
> **Ours**|**55.52**|**76.23**|**65.88**|**53.81**|83.47|91.11|**98.22**|81.65
>
> Refs:
>
> [1] Jiang et al. CaLa: Complementary Association Learning for Augmenting Composed Image Retrieval. SIGIR 2024.
>
> [2] Bai et al. Sentence-level Prompts Benefit Composed Image Retrieval. ICLR 2024.
>
> [3] Tang et al. Modeling Uncertainty in Composed Image Retrieval via Probabilistic Embeddings. ACL (1) 2025.
>
> [4] Tian et al. CCIN: Compositional Conflict Identification and Neutralization for Composed Image Retrieval. CVPR 2025.
>
> We sincerely appreciate the time and effort in reviewing our manuscript.

---

### Official Review · Reviewer_QsfS · 2025-07-02

**Clarity:** 2
**Significance:** 2
**Originality:** 2
**Rating:** 4
**Confidence:** 4

**Summary:**

This paper introduces Robust Uncertainty Calibration (RUNC), a novel approach designed to improve composed image retrieval (CIR) by explicitly modeling semantic uncertainty in matching bi-modal queries to target images. RUNC quantifies uncertainty by using a Normal-Inverse Gamma (NIG) prior distribution to model the semantic covariances among candidate images. This estimated uncertainty is then used to calibrate the training objective, promoting a more balanced semantic alignment by focusing on ambiguous and rare concepts. To further enhance the query representation, RUNC incorporates an Implicit Concept Matching (ICM) module that reduces redundancy by enforcing orthogonality between textual and implicit visual components. Experiments conducted on the benchmark FashionIQ and CIRR datasets confirm that RUNC achieves robust and reliable performance, outperforming previous state-of-the-art methods.

**Questions:**

1. The connection between the Implicit Concept Matching (ICM) module and the paper's central theme of uncertainty should be clarified.
2. To allow for a complete assessment of the method's practical trade-offs, the paper should include a quantitative analysis of the comprehensive computational overhead. A table comparing training time or GFLOPs against multiple baselines would enable a better evaluation of the model's efficiency.
3. The qualitative analysis requires expansion beyond the single success case shown in Figure 4.

**Ethical Concerns:**

["NO or VERY MINOR ethics concerns only"]

**Final Justification:**

After carefully reading the rebuttal of the authors, most of my concerns have been addressed. Therefore, I raise my rating to borderline accept.

**Limitations:**

yes

**Quality:**

2

**Strengths And Weaknesses:**

Strengths
1.	The paper introduces a novel framework for Composed Image Retrieval (CIR) that explicitly quantifies both aleatoric (data noise) and epistemic (model overconfidence) uncertainty using a Normal-Inverse Gamma (NIG) distribution to model semantic correlations.
2.	The method uses the estimated uncertainty to create a balanced contrastive loss, which adaptively increases the penalty on ambiguous or rare samples, mitigating the influence of dominant semantics and improving learning from difficult examples.
3.	RUNC achieves robust and competitive results on two widely-used benchmark datasets, FashionIQ and CIRR, outperforming previous advanced methods.
Weaknesses
1. Implicit Concept Matching (ICM) appears only loosely connected to the paper’s central theme of uncertainty modeling. Although the authors claim it alleviates fusion ambiguity, it is unclear whether this ambiguity truly maps onto the aleatoric or epistemic uncertainty defined in the paper. ICM seems better characterized as an independent module for refining fusion representations rather than as an integral part of the uncertainty‑calibration pipeline. While beneficial, its role within the Uncertainty Calibration should be articulated more explicitly and positioned more cautiously.
2. The calculation of the semantic covariance matrix requires computing pairwise cosine similarities between the features of all candidate samples within each batch. This operation could become computationally expensive and may impact training efficiency, particularly when using a large batch size.
3. The clarity of the framework diagram in Figure 2 could be improved. Several key symbols, such as p, p*, and q* may be confusing for the reader. The inclusion of a formal legend or more descriptive labels to define these variables would significantly enhance the diagram's readability.
4. Figure 4 only shows one successful retrieval case to intuitively demonstrate the uncertainty quantification of the model. Analyzing the instances where the model fails and their corresponding uncertainty scores will provide a deeper understanding of the limitations of the model and help verify whether the quantified uncertainty is truly effective in challenging scenarios.

---

> ### Author Rebuttal · Authors · 2025-07-31
>
> ***Q1/W1: Clarify the connection of between the Implicit Concept Matching (ICM) module and uncertainty.***
>
> **A1:** Thank you for providing this suggestion.  We are sorry for the confusion and would like to clarify the ICM module as follows:
> - In our design, ICM is **not an independent feature fusion module**, but a mechanism specifically aimed at mitigating the uncertainty, particularly the aleatoric uncertainty, in the composition of reference images and modifications.  As the reference images usually have contradictory semantics with modifications (the modified visual regions), partially alignments, and redundant visual information (e.g.. model poses, lighting, irrelevant backgrounds), the fusion queries $\boldsymbol{q}$ generated  from reference images and modification texts are prone to possessing unstable representations and inconsistent semantics as the users desired. To this end, we introduce an implicit query $\boldsymbol{p}^\prime$ to directly guide the learnable prompt $\boldsymbol{p}$ during fusion query generation, to facilitate the learning of more explicit semantics with direct alignments to the target images.  From the perspective of similarity measurement in the retrieval task, the ICM module **acts as a front-end to eliminate uncertainty** and ensure the semantic consistency of inputs (i.e., query fusion), the remaining parts of the proposed RUNC could be regarded as back-end constraints to avoid over-confidence of output (i.e., matching scores). They collaborate to minimize uncertainties in the composed image retrieval task.
> - To verify the connections between ICM and uncertainty, we have implemented ablative experiments to present the trends of aleatoric uncertainty scores during training as follows. The model without the ICM module generally **has large uncertainty scores** compared to our proposal and **struggles with training convergence**, which demonstrates that ICM contributes to the reduction of aleatoric uncertainty. Without the guidance of implicit queries, the inconsistent and low quality of hybrid-modal inputs consistently causes confusion when matching with candidate samples.
>
> Models|Epoch 0|Epoch 5|Epoch 10|Epoch 15|Epoch 20
> ----|----|----|----|----|----
> w/o ICM|0.902|0.614|0.334|0.199|0.209
> w ICM|0.692|0.234|0.118|0.086|0.086
>
> - Besides, to provide a vivid presentation of how ICM works, **Figure 8 in supplementary materials** shows heatmaps of the prompt embedding $\boldsymbol{p}$ and text features $\boldsymbol{m}$ after the ICM module, where $\boldsymbol{p}$ highlights the maintained regions in references, and $\boldsymbol{m}$ specifies the correlated modified areas. Additionally, Table 3 demonstrates that the ICM module brings progressive improvements.
>
> To sum up, the ICM module concentrates on reducing the aleatoric uncertainty in query compositions and **collaborates with subsequent designs as the early purification of features** to jointly ensure the reliability of the predicted matching scores. Quantitative and visualization results also demonstrate the effectiveness of the ICM module.
>
> ***Q2/W2: Computational overhead, especially concerns about the covariance matrix.***
>
> **A2:** Thank you for your insightful comment.  We appreciate the reviewer’s concern regarding the computational cost of the semantic covariance matrix in each batch.
> - In practical training, our method has been evaluated **under moderate batch sizes** (within 256, comparable to the current mainstream methods) in a single NVIDIA A800 GPU. And the additional overhead is manageable and does not cause significant slowdowns in training, as seen in the table below.
> - Moreover, we emphasize that the target features used in this computation are **readily accessible and already indexed in the batch-wise memory**, as they are shared across covariance computations and semantic alignments. This enables the pairwise cosine similarity computation to be performed using efficient matrix operations without additional forward passes or repeated encoding steps.
> - In our experiments, we observed that the proposed RUNC incurs 2.6G FLOPS compared to a baseline model (BLIP2-ViT/G). Besides, we compare the training time and inference time as shown in the table below. Compared with CaLa[1] with complex twin compositors and multiple operations on target features, our model exhibits low computational complexity during training and **demonstrates high conciseness and efficiency in the reasoning phase**, as the matching scores can be directly calculated from the cosine distance between the query fusion feature and the candidate features.
>
> Models | Training Time(s) | Inference Time(s)
> ----  | ----  |----
> CaLa[1] | 263.1 | 45.7
> SPRC[2] | 209.5 | 27.2
> Ours  | 214.3 | 27.4
>
> - Furthermore, given the improvement of incorporating semantic covariance in boosting the retrieval performances (as shown in Table 3 in the main paper), the semantic covariance and uncertainty estimation in our methods consistently improve recall rates across multiple benchmarks. We believe this constitutes a **reasonable trade-off** between computational cost and retrieval performance, especially in settings where robustness to semantic ambiguity is critical. We would include a discussion of computational challenges when scaling to very large batch sizes in the revised manuscript. In future work, we also consider exploring efficient memory bank sampling or low-rank projections to further improve scalability.
>
> ***Q3/W4: Analysis of failure cases in addition to a single success case depicted in Figure 4.***
>
> **A3:** Thank you for raising this important concern. Due to the length limit of the main manuscript, we have added **failure cases in FashionIQ and CIRR in Figure 11 and 12 in the supplementary materials**. Specifically, for the second failure case in Figure 11, the uncertainty scores for the top-10 instances are all over 0.34, which could be caused by ambiguous queries (“attractive” in the query text is too vague to identify the targets) and semantic overlap (all the top-10 images embody characteristics of “black”, “bold”, “dress” that are very visually similar to the target one) in these instances. We appreciate your suggestions and would add failure cases in the final version with uncertainty scores and analysis.
>
> ***W3: The clarity of the framework diagram in Figure 2 could be improved***
>
> **A4:** Thanks for your suggestions. Specifically, $\boldsymbol{p}$ means learnable prompts, and $\boldsymbol{p}^\ast$ means virtual guidance that maintains the identical shape as $\boldsymbol{p}$. $\boldsymbol{q}$ is the fusion query, and $\boldsymbol{q}^\ast$ refers to the implicit query that keeps the same shape as $\boldsymbol{q}$. These two sets of symbols appear in pairs, with the asterisk superscripted features ($\boldsymbol{p}^\ast$, $\boldsymbol{q}^\ast$) guiding the semantic expression of the original features ($\boldsymbol{p}$, $\boldsymbol{q}$) in ICM module. We would add legends for key symbols in the revised version to improve the diagram's readability.
>
> Refs:
>
> [1] Xintong Jiang, Yaxiong Wang, Mengjian Li, Yujiao Wu, Bingwen Hu, Xueming Qian: CaLa: Complementary Association Learning for Augmenting Composed Image Retrieval. SIGIR 2024.
>
> [2] Yang Bai, Xinxing Xu, Yong Liu, Salman Khan, Fahad Khan, Wangmeng Zuo, Rick Siow Mong Goh, Chun-Mei Feng: Sentence-level Prompts Benefit Composed Image Retrieval. ICLR 2024.
>
> We sincerely appreciate your valuable suggestions and recognition of our work, which have greatly contributed to the improvement of our manuscript.

---

### Comment · Area_Chair_Ep2W · 2025-08-06
**Reminder of Submitting Final Score**

Dear Reviewers,

This is a gentle reminder that the authors added their response on your question. Can you provide your feedback on their response? Please keep in mind that the deadline of August 6th approaching, and your additional timely feedback greatly enhance further discussions if needed.

Best regards
AC

---

### Note · Authors · 2025-08-13

We thank all reviewers for valuable feedback and appreciate the Chairs for the time and attention to our submission. We are encouraged by the recognition as:
- **Practical novel framework** (*gEs2, gQCa, K9V5, QsfS*)
- **Theoretical rigor & clarity** (*gQCa, K9V5*)
- **Competitive performance** (*gEs2,QsfS*)

The most common concerns with responses are:
- ***Clarity of Implicit Concept Matching (ICM)*** (*K9V5, QsfS*): ICM aims at mitigating uncertainty in composing modifications and reference images, which often contain semantics conflicting with modifications, partial alignments, and redundant information, leading to unstable fusion queries. We introduce an implicit query $\boldsymbol{p}^\prime$ to guide the learnable prompt $\boldsymbol{p}$ via explicit alignment with targets.  ICM is **not an independent fusion module but collaborates with uncertainty calibration module as an early purification step** to ensure reliable matching scores. Ablation (Tab. 3) and uncertainty trends confirm its role in reducing uncertainty.
- ***Computation cost*** (*K9V5, QsfS*): RUNC requires 2.6G FLOPS versus BLIP2-ViT baseline. Training/validation time is 214.3/27.4s, faster than CaLa (263.1/45.7s) and comparable to SPRC (209.5/27.2s). During inference, pre-indexing frozen encoder features and disabling ICM reduce runtime. All experiments converged within 4 hours on one A800 GPU, **showing efficiency comparable to SOTA**.
- ***Interpretability of learnable prompts*** (*gEs2, gQCa*): The learnable prompts, derived from \<image embeddings, text tokens, and learnable queries\>, enhance visual **semantics preserved in the reference images** (e.g., outlines, objects, reversed attributes). Due to partial alignment and weak supervision, prompts inherently contain unclear semantics, motivating ICM to reduce ambiguity in query fusion. **Fig. 8 (in Supp.) attention maps** show ICM decouples retained and modified regions, distilling retained visual areas into learnable prompts.

We would polish our paper in the revised version:
- Highlight the connection of ICM and uncertainty (Intro, Methods)
- Include computation efficiency analysis (Exp.)
- Provide ICM structure/Algo (Appendix)
- Move visualization and failure cases to main paper
- Add legends in Fig.2 & fix typos in Eq.1, 8, 11

Finally, we extend sincere appreciation to the Chairs and reviewers for improving the quality of our manuscript. We hope these clarifications and additions will meet the high standards of NeurIPS.

---

### Decision · Program_Chairs · 2025-09-17

**Decision:**

Accept (poster)

**Comment:**

This paper proposes RUNC (Robust UNcertainty Calibration), a novel framework for composed image retrieval (CIR) that explicitly models both aleatoric and epistemic uncertainty in matching bi-modal queries (reference image + modification text) to target images. RUNC employs a high-order Normal-Inverse Gamma (NIG) distribution to estimate semantic covariance among candidate images and uses this uncertainty to calibrate the training objective, promoting balanced semantic alignment. To further refine query representation, an Implicit Concept Matching (ICM) module enforces orthogonality between explicit textual embeddings and implicit visual cues. Experiments on FashionIQ and CIRR datasets demonstrate that RUNC achieves robust and reliable retrieval, outperforming prior state-of-the-art methods. Reviewers note that the framework is technically solid, theoretically motivated, and demonstrates meaningful contributions to multi-modal retrieval. Weaknesses include limited clarity in the connection of ICM to uncertainty calibration, computational overhead of pairwise covariance calculations, and some missing details about module architecture and mathematical assumptions. Despite these, the authors provide thorough ablations, qualitative analyses, and rebuttal clarifications that address key concerns.